# Distributed Knowledge Subspace Hypothesis: Evidence from Generalization Failure in Parametric Knowledge Editing

## Abstract

Large language models (LLMs) store factual knowledge at scale, and recent editing methods allow local updates of target facts. While effective in single prompt styles, prior evaluations have overlooked whether edits generalize across task formats. In this work, we present a systematic study of cross-format generalization in parametric knowledge editing. Across six curated task formats, we show that edits that succeed in the source format often fail to transfer, and high frequency in training does not guarantee a shared, format-invariant knowledge structure. Representation-level analyses reveal that edit directions cluster by task format, supporting a *distributed knowledge subspace hypothesis* in which knowledge is distributed across multiple format-specific subspaces. To mitigate these failures, we introduce multi-format supervision with iterative expansion, which improves transferability with minimal overhead. Our study reframes generalization failure not only as a practical limitation of current editing methods but also as evidence about the distributed memory mechanisms underlying factual knowledge in language models.

## 1 Introduction

Knowledge editing methods aim to update specific facts stored in large language models (LLMs) (Fang et al., 2025; Dong et al., 2025; Cohen et al., 2024; Zeng et al., 2024; Li et al., 2024; Zheng et al., 2023; Hartvigsen et al., 2023; Zhong et al., 2023; Meng et al., 2023; 2022; Mitchell et al., 2022b;a; Dai et al., 2022; De Cao et al., 2021; Zhu et al., 2020). While prior evaluations of these methods span a range of dimensions (Cohen et al., 2024; Rosati et al., 2024; Zeng et al., 2024; Zhang et al., 2024; Zhong et al., 2023; Meng et al., 2022; Levy et al., 2017), their ability to generalize across task formats remains underexplored.

In this work, we show that current parametric editing methods exhibit systematic generalization failures. Edits that succeed in completion-style prompts often fail in other task formats, such as multiple-choice and true/false questions. Our findings also show that even highly frequent facts do not necessarily induce a generalized structure across formats. Furthermore, representation-level analyses reveal that edit directions diverge by task format, suggesting the *distributed knowledge subspace hypothesis*: different task formats engage distinct representational subspaces rather than converging on a single unified representation for the fact. This perspective reframes generalization failure not merely as a limitation of current editing methods but as empirical evidence about the memory mechanisms of LLMs.

Our contributions are threefold. First, we introduce an evaluation framework for knowledge editing across diverse task formats. Second, we show that high frequency in training does not induce a shared knowledge structure across formats, and provide representation-level evidence for the distributed knowledge subspace hypothesis. Third, we explore multi-format supervision with iterative expansion as a promising step toward robust editing.

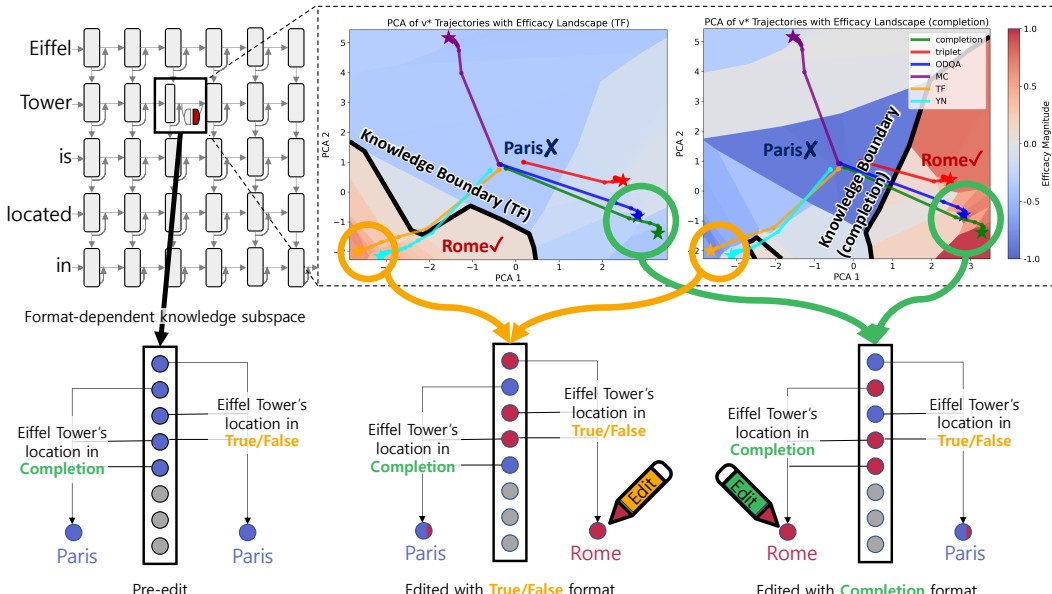

Figure 1: **Divergent edit dynamics across formats.** For the target edit *(Eiffel Tower, location, Paris ⇒ Rome)*, we project MEMIT's value optimization process into the local representation space. Each trajectory shows how the edit vector moves from its pre-edit state $v^0$ toward the optimized target $v^*$ under a specific format. We find that TF and YN formats follow partially aligned paths, while others diverge into distinct regions. Background heatmaps highlight regions of successful (red) and failed (blue) edits in True/False QA (left) and completion (right), revealing that knowledge transition boundaries are rotated and shifted across task formats. Together, these patterns illustrate why an edit that works in one format often fails to generalize reliably to others.

## 2 EVALUATION FRAMEWORK FOR KNOWLEDGE EDITING ACROSS FORMATS

We define a knowledge edit as a tuple $(s, r, o^c \Rightarrow o^*)$, where $s$ is a subject entity, $r$ is a relation, and the current object $o^c$ is updated to a new object $o^*$. To evaluate cross-format generalization, we instantiate this edit tuple across a set of prompt formats, thereby testing whether edits to a fact applied in one format remain valid in others. We distinguish between a *source format* $f_{src}$, in which the edit is applied, and a set of *target formats* $\mathcal{F}_{tgt} = \{f_1, \ldots, f_k\}$, on which the edit is evaluated.

**Format Variation Design.** We consider six curated formats: **Completion** as a free-form continuation prompting the missing object directly, **Triplet** as a structured schema $(s, r, \cdot)$ that explicitly exposes the relational slot, **Open-Domain QA (ODQA)** as a natural question template phrased in interrogative form, **Multiple-Choice QA (MC)** as a closed-choice format where $o^c$ and $o^*$ are mapped to categorical labels, **True/False QA (TF)** as a binary statement verification template evaluating the truth of a factual assertion, and **Yes/No QA (YN)** as a binary question template asking whether the subject $s$ is associated with a candidate object. Together, these formats provide broad coverage of prompt-response styles observed in downstream applications. Detailed templates and instantiations for each format are provided in Appendix A.

**Metrics.** We evaluate edits using two complementary metrics (Meng et al., 2022). **Efficacy** is defined as $\mathbb{1}[\mathbb{P}[o^*|s, r] > \mathbb{P}[o^c|s, r]]$, indicating whether the model prefers the edited object over the original one. **Specificity** is defined as $\mathbb{1}[\mathbb{P}[o^c|s_n, r] > \mathbb{P}[o^*|s_n, r]]$ for a neighborhood subject $s_n$ that shares the same original object ($o^c$), measuring whether unedited knowledge is preserved. We report the average specificity across a set of ten neighborhood subjects for each instance. Together, these metrics quantify both the success of the intended edit and unintended side effects when evaluated across formats.

## 3 EXPERIMENTS

### 3.1 EXPERIMENTAL SETUP

**Editing Methods.** We compare three parametric editing methods against an unedited model. The **Baseline** corresponds to the unedited model, serving as a reference point for pre-existing behavior. ROME (Meng et al., 2022) injects new knowledge via targeted weight updates with knowledge localization at a single layer. **MEMIT** (Meng et al., 2023) extends ROME to enable batch edits for multiple facts with multi-layer update. AlphaEdit (Fang et al., 2025) builds on MEMIT by incorporating null-space constraints that penalize deviations from the original distribution. In addition, we test whether generalization failures also arise in non-parametric editing. Specifically, **in-context knowledge editing (IKE)** (Zheng et al., 2023) is a non-parametric baseline in which knowledge update demonstrations are provided in the input context. We use MEMIT for our main experiments. Implementation details are provided in Appendix B.

**Datasets.** We conduct experiments on **CounterFact-1k**, a subset of 1,000 triples from the CounterFact benchmark (Meng et al., 2022). Additionally, we evaluate on 1,218 triples from the **Physical Event Plausibility (PEP)** dataset (Porada et al., 2021), to examine whether our findings extend beyond encyclopedic factual knowledge. For PEP, we construct edit instances by grouping triples that share the same subject and relation, and treating the plausible object as $o^c$ and the implausible object as the target object $o^*$, yielding all possible edit pairs for evaluation. To systematically evaluate cross-format behavior, we instantiate every fact in the six formats introduced in Section 2.

**Models.** Our main experiments are conducted on **Llama-3.2-3B-Instruct** (Dubey et al., 2024), an open-source instruction-tuned language model. To assess the generality of our findings, we evaluate a broader set of models: **Llama-3.2-1B-Instruct**, **Llama-3.1-8B-Instruct**, **OLMo2-Instruct (1B, 7B, 13B)** (Walsh et al., 2025) and **Qwen3 (0.6B, 1.7B, 4B, 8B, 14B)** (Yang et al., 2025).

### 3.2 CROSS-FORMAT GENERALIZATION FAILURE

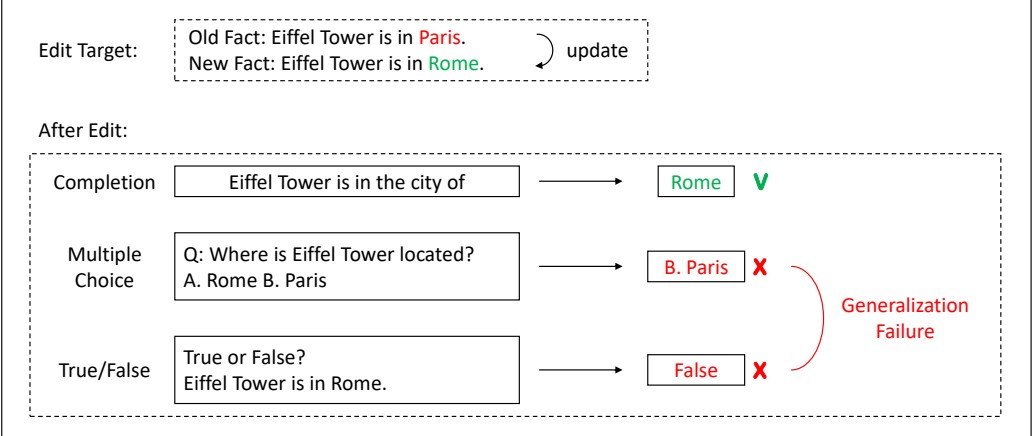

Figure 2: A conceptual illustration of generalization failure across task formats. Although the location of Eiffel Tower is updated to Rome in the completion format, the update fails to generalize to other formats.

We first evaluate whether knowledge edits performed in the *completion* format generalize to alternative formats. Figure 3 reveals clear limitations in cross-format generalization on CounterFact-1k. All editing methods achieve strong efficacy on completion prompts ($> 0.9$), confirming that the injected knowledge is correctly retrieved in the source format. However, transfer performance drops sharply in other formats. For instance, MEMIT retains only moderate efficacy on triplet and ODQA (0.71 and 0.77) and degrades further on MC, TF and YN (down to 0.43). These results indicate that success in the edited format is not a reliable indicator of robustness across diverse task formats.

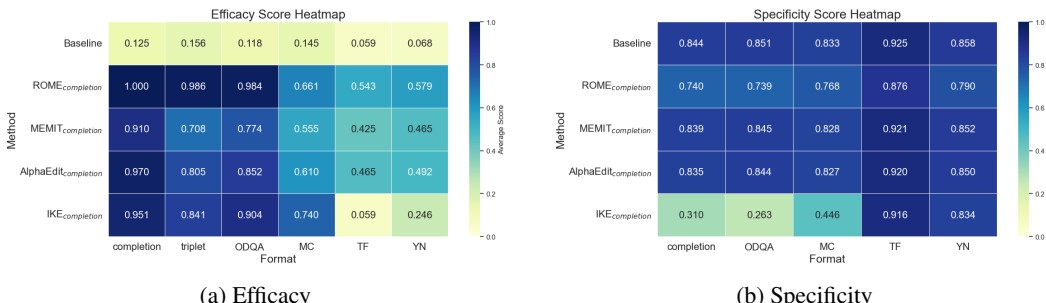

(a) Efficacy                (b) Specificity

Figure 3: Cross-format generalization results on Llama-3.2-3B-Instruct and CounterFact-1k when edits are performed in the completion format. Each cell reports the average score across all samples for each format. **Efficacy:** All editing methods achieve high efficacy in the source format (completion) but show severe degradation when transferred to other formats, specifically in TF and YN formats. **Specificity:** Parametric editing methods tend to preserve specificity, while IKE suffers from strong efficacy-specificity trade-offs.

Specificity analysis further highlights this limitation. Parametric editing methods tend to preserve specificity at levels comparable to the baseline, showing that weight edits remain relatively well localized. In contrast, IKE suffers from substantial side effects: specificity scores fall drastically in completion (0.31) and ODQA (0.26), indicating that contextual injection destabilizes unrelated knowledge. Taken together, these findings highlight systematic failures of cross-format generalization, pointing to deeper challenges in editing and evaluating knowledge of language models.

**Generalization Patterns Across Models and Datasets.** To assess whether cross-format fragmentation is specific to a particular model, we extend our evaluation to a broad set of models spanning three model families (Llama, OLMo2, Qwen3) and parameter ranges from 0.6B to 14B. Using MEMIT$_{completion}$ on CounterFact-1k, we observe that larger models generally exhibit smaller cross-format variance, but even the largest models tested (14B) retain a sizeable gap. The smallest cross-format gap, achieved by Llama-3.1-8B-Instruct, is still 0.25 between its maximum and minimum efficacy.

We further investigate whether this behavior generalizes to other types of relations, we examine the PEP dataset, a benchmark for commonsense physical reasoning. Using Llama-3.2-3B-Instruct and MEMIT$_{completion}$, we observe the same pattern of cross-format fragmentation: editing only with the completion format yields an efficacy of $0.38 \pm 0.18$ (min = 0.16, max = 0.70).

The consistent presence of cross-format discrepancies across diverse models and datasets demonstrates that the phenomenon is not confined to any specific model or dataset. Since our investigation is computationally limited to a specific set of models and datasets, further exploration remains an important direction. Full results are provided in Appendix C.

### 3.3 PRE- VS. POST-EDIT CONSISTENCY

To examine the effects of parametric knowledge editing to cross-format generalization failures, we analyze *pre-* and *post-edit consistency* across formats on CounterFact-1k with Llama-3.2-3B-Instruct. Consistency is measured using an average pairwise metric (Elazar et al., 2021): for each instance, we compute the proportion of format pairs that produce semantically aligned predictions. Overall results show a sharp decline in consistency after editing with MEMIT. The baseline model exhibits a high average consistency of 0.86, reflecting stable responses across different task formats. In contrast, MEMIT drops consistency to 0.66. Importantly, this suggests that being consistent pre-edit is not a guarantee of robustness after editing.

Prior work suggest that the frequency of subject-object pairs in the training corpus strongly influences a model's ability to recall factual associations (Kang & Choi, 2023; Kandpal et al., 2023; Elazar et al., 2022; Li et al., 2022). To examine whether this factor explains cross-format consistency, we group factual triples by their estimated frequency and compare pre- and post-edit behavior. We leverage the Infini-gram framework (Liu et al., 2024) to estimate subject–object frequencies in

Table 1: Pre- and post-edit average consistency on CounterFact-1k across corpus frequency bins.

| Frequency bin | Baseline | MEMIT$_{\text{completion}}$ | Number of samples |
|:---:|:---:|:---:|:---:|
| $x = 0$ | 0.83 | 0.67 | 577 |
| $0 < x \leq 10$ | 0.74 | 0.68 | 21 |
| $10 < x \leq 100$ | 0.85 | 0.66 | 74 |
| $100 < x \leq 1000$ | 0.91 | 0.64 | 194 |
| $1000 < x \leq 10000$ | 0.96 | 0.60 | 95 |
| $x > 10000$ | 0.96 | 0.63 | 39 |

the RedPajama corpus (Weber et al., 2024). Table 1 reports average consistency within frequency bins. Before editing, corpus frequency is a strong predictor of consistency: rare facts ($x = 0$) achieve 0.83, while very frequent facts ($x > 10000$) reach 0.96. This aligns with the intuition that repeated exposure during training reinforces robust retrieval across input formats. After editing with MEMIT, however, the model exhibits uniformly low consistency across all bins, with no clear correlation. This suggests that high frequency in training may encourage the model to form multiple representational subspaces rather than a single abstract, format-invariant representation of the same fact.

## 3.4 REPRESENTATION-LEVEL ANALYSIS

### 3.4.1 BACKGROUND ON MEMIT

Following prior work (Meng et al., 2023; 2022), we view $W_{proj}^l$ as a linear associative memory (Kohonen, 1972; Anderson, 1972), where $W_{proj}^l$ is the MLP projection matrix at layer $l$. This perspective suggests that any linear operation $W$ can associate a set of vector keys $K = [k_1 \mid k_2 \mid \ldots]$ (encoding subject entities) and corresponding values $V = [v_1 \mid v_2 \mid \ldots]$ (encoding attributes) by minimizing the squared error $\|\hat{W}K - V\|$. A set of new associations $(K_1, V_1)$ can then be inserted by solving a constrained least-squares problem, leading to the closed-form update:

$$\hat{W} = W_0 + RK_1^\top \left(C_0 + K_1 K_1^\top\right)^{-1}, \tag{1}$$

where $K_1$ contains the keys for edited subjects, $R \triangleq V_1 - W_0 K_1$ stacks the residuals of their desired value shifts, and $C_0 = \lambda \cdot \mathbb{E}_k[kk^\top]$ is the pre-computed covariance statistic of original keys sampled from text, scaled by a hyperparameter $\lambda$ that regularizes the weighting of new vs. existing associations.

The editing procedure requires selecting appropriate key and value vectors to insert new associations. To obtain a key vector $k^*$, we sample multiple prefix contexts ending with the subject token and average their MLP input representations. The target value $v^*$ (MLP output) is then optimized such that inserting $(k^*, v^*)$ shifts the model's prediction from the original object $o^c$ to the new object $o^*$. This optimization combines a loss encouraging high probability of $o^*$ with a regularization term that preserves the model's representation of the subject, thereby controlling for essence drift. The resulting $(k^*, v^*)$ pairs are directly inserted into the projection matrix update in Eqn. 1.

### 3.4.2 FORMAT-SPECIFIC TRAJECTORIES AND SUBSPACES

To further investigate why edits fail to generalize across formats, we analyze representation-level changes induced by MEMIT updates. In particular, we track the optimization trajectories [1] of the target value vectors $v^*$, which encode the shift from the original object $o^c$ to the new object $o^*$. By comparing these trajectories and the geometry of the resulting edit directions across different formats, we obtain a representational view of how knowledge edits diverge into distinct subspaces rather than converging to a unified representation.

Figure 1 visualizes the optimization trajectories $(v_f^0, \ldots, v_f^*)$ for each format $f$ in a two-dimensional PCA projection (Pearson, 1901) for a curated example. Each trajectory illustrates how the gradient-based optimization progresses toward the format-specific target vector $v_f^*$. We observe that different

---

[1] Although our analysis focuses on MEMIT, this value update step is shared across ROME and AlphaEdit.

formats follow diverging paths: while formats with similar surface structures (e.g., TF, YN) move in partially aligned directions, others are driven into separate regions. Background heatmaps show the edit-efficacy magnitude, defined as $\mathbb{P}_{v^i}[o^*] - \mathbb{P}_{v^i}[o^c]$ for candidate vectors $v^i$ along the trajectories, with interpolation across neighboring points; higher values are shown in red and lower values in blue. The contrast between True/False QA (left) and completion (right) demonstrates that edits which are successful in one format may not cross the decision boundaries of another.

We next examine the geometry of the final edit directions $(v_f^* - v_f^0)$ across all instances. A 2D PCA projection (Appendix D) reveals clear clustering by format, indicating that knowledge edits are not stored in a unified subspace but instead distributed across multiple, format-dependent regions. To quantify this separability, we train a linear probe to predict the format from edit directions. Using 6,000 vectors (1,000 facts × 6 formats) with a 50:50 train–test split, the probe achieves 98.4% test accuracy using only the top 50 principal components. This near-perfect classification confirms that edit directions carry distinct, format-specific signatures rather than converging to a common representation of the same underlying fact.

Together, these findings suggest that factual knowledge in language models is distributed across multiple format-dependent subspaces. Edits that align with one subspace may fail to propagate to others, offering a mechanistic explanation for the cross-format generalization failures observed at the behavioral level.

## 4 Toward Robust Parametric Knowledge Editing

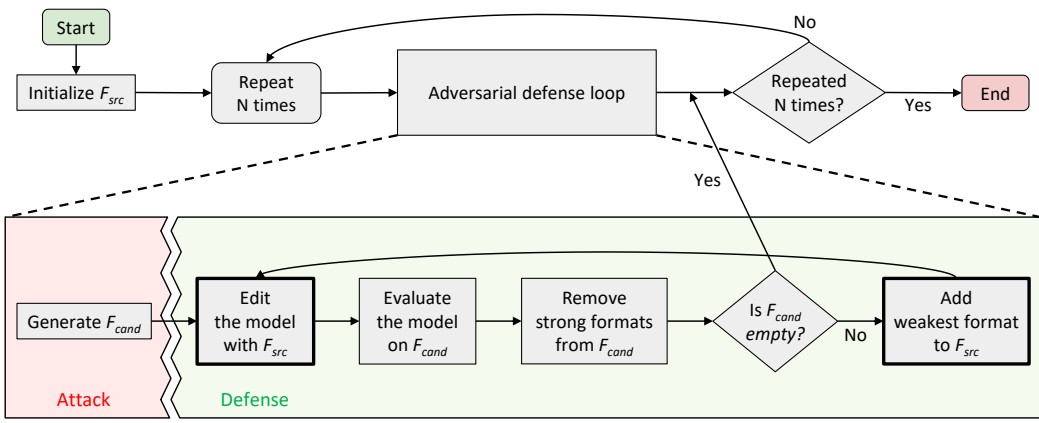

Figure 4: A flowchart of the adversarial defense loop. Each iteration consists of an attack phase, which generates candidate formats $\mathcal{F}_{cand}$ to probe the model, and a defense phase, which edits the model with $\mathcal{F}_{src}$, evaluates on candidates, and adds the weakest format to $\mathcal{F}_{src}$. Iterative expansion continues until no vulnerable formats remain.

The analyses in Section 3 show that current editing methods do not generalize reliably across task formats, reflecting the distributed nature of factual knowledge in language models. To address this, we propose an *adversarial defense loop* (Figure 4) that incrementally identifies and patches format-level vulnerabilities. Each iteration of the loop consists of an *attack* phase, where candidate formats $\mathcal{F}_{cand}$ are generated to probe the model, and a *defense* phase, where the vulnerable formats are added to the supervision set $\mathcal{F}_{src}$ and the model is edited using multiple formats in this set. Repeating this process progressively strengthens robustness against unseen formats.

### 4.1 Adversarial Defense: Multi-Format Supervision with Iterative Expansion

For the defense phase, our core method leverages *multi-format supervision*, where each factual edit is instantiated across multiple prompt formats. A naive extension of single-format editing would jointly optimize multiple formats (*multi-key + multi-value*), treating each format as an independent factual edit under a batch update scheme. However, recent analyses of parametric editing show that optimizing different value vectors with near-identical keys (from the same subject entity) can induce

update conflicts during editing (Dong et al., 2025). Therefore, we adopt a *multi-key + joint-value* formulation that encourages all formats to agree on a single, format-invariant edit direction and reduces fragmentation across subspaces. Concretely, for a target edit $(s, r, o^c \rightarrow o^*)$ and a source set of formats $\mathcal{F}_{src}$, we optimize a shared value vector using a joint objective inspired by Dong et al. (2025):

$$v^* \;=\; \arg\min_v \sum_{f \in \mathcal{F}_{src}} - \log P_v[o_f^* \mid s, r, f], \tag{2}$$

which aligns the update direction across all supervised formats.

To minimize overhead, we adopt an *iterative expansion* procedure that follows the adversarial defense loop in Figure 4. We begin with $\mathcal{F}_{src} = \{f_{completion}\}$ and initialize the candidate pool $\mathcal{F}_{cand}$ with the six human-crafted formats introduced in Section 2. At each iteration, the loop: (1) edits the model with the formats in $\mathcal{F}_{src}$; (2) evaluates the edited model on all formats in $\mathcal{F}_{cand}$; (3) removes strong formats (those with efficacy $> \tau$, with $\tau = 0.9$) from $\mathcal{F}_{cand}$; (4) if $\mathcal{F}_{cand}$ is non-empty, adds the weakest format (the one with lowest efficacy) to $\mathcal{F}_{src}$; otherwise, terminates. This iterative expansion progressively strengthens robustness by defending against the most vulnerable formats while avoiding the need to supervise every possible format.

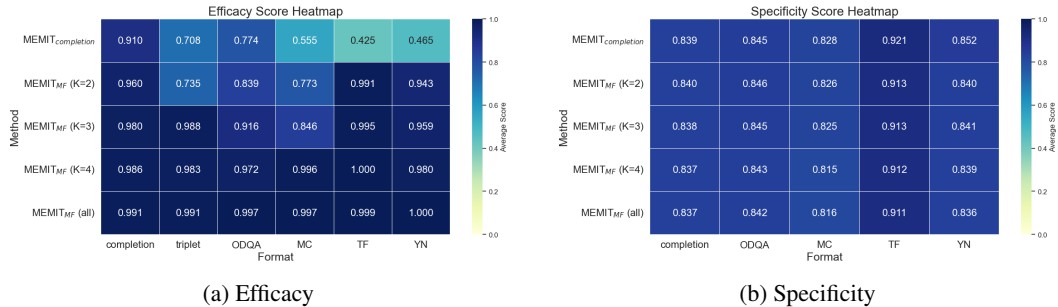

(a) Efficacy             (b) Specificity

Figure 5: Cross-format generalization results under multi-format supervision with iterative expansion on Llama-3.2-3B-Instruct and CounterFact-1k. The supervision set (source formats) is progressively expanded (completion $\rightarrow$ TF $\rightarrow$ triplet $\rightarrow$ MC), and terminates at $K = 4$ when no vulnerable formats remain. The resulting performance nearly matches that of supervising on all formats, achieving $> 0.97$ efficacy across tasks while specificity remains stable throughout expansion.

**Main Results.** Figure 5 reports efficacy and specificity under multi-format supervision with iterative expansion on Llama-3.2-3B-Instruct and CounterFact-1k. Starting from a single completion format, the loop first incorporates TF, then triplet, and finally MC before terminating at $K = 4$, since no remaining candidate formats fall below the vulnerability threshold ($\tau = 0.9$). Efficacy improves consistently as the supervision set expands, and with only four supervised formats (MEMIT$_{MF}$ ($K = 4$)) performance nearly matches that of the full supervision condition (MEMIT$_{MF}$ (all)), achieving near-perfect efficacy ($> 0.97$) across all formats. This shows that multi-format supervision effectively counteracts the fragmentation of knowledge representations, and that iterative expansion achieves these gains efficiently without supervision over every format. Specificity scores remain stable throughout the process (Figure 5b), comparable to the single-format baseline. Thus, gains in cross-format efficacy do not come at the cost of increased collateral edits, preserving the locality of updates.

**Ablation: Effect of Joint Loss.** We additionally test whether joint optimization is necessary by applying six edits simultaneously without the joint loss. This corresponds to treating each format as an independent edit under a batch update scheme. The resulting efficacy is $0.887 \pm 0.044$ (min $= 0.827$, max $= 0.932$), substantially below the $0.996 \pm 0.004$ achieved with our joint loss. This empirically confirms that independent updates for formats sharing near-identical keys interfere with each other, whereas joint optimization encourages alignment across formats.

**Ablation: Disentangling Key Variation and Value Divergence.** Although our analyses focus on divergence in value space, key variation may also play an important role as different formats naturally yield different keys. To isolate the contributions of key variation and value divergence, we

conduct two targeted ablations: (1) *multi-key + single-value* and (2) *single-key + joint-value* editing. In both cases, the completion format provides the single key or single value. The proposed method (*multi-key + joint-value*) achieves $0.996 \pm 0.004$, when edited with the six formats. When restricting the update to a single key while keeping the joint value (single-key + joint-value), efficacy drops sharply to $0.800 \pm 0.065$ (min = 0.686, max = 0.861). Conversely, when allowing key variation but optimizing only a single value (multi-key + single-value), performance deteriorates to $0.788 \pm 0.213$ (min = 0.516, max = 0.997). These ablations demonstrate that *neither key variation nor value divergence alone* is sufficient to explain or correct cross-format fragmentation. Both components contribute meaningfully: different formats induce distinct keys, but their corresponding value updates also diverge. The combination of multi-key access patterns and a shared value optimization is therefore essential for achieving robust cross-format generalization.

**Ablation: Order Sensitivity in Iterative Expansion.** We also examine whether the order in which formats are added to the defense set affects robustness or efficiency. Instead of selecting the lowest-scoring candidate format at each iteration (our default strategy), we reverse the selection rule and choose the highest-scoring remaining format. With the reversed order, the model no longer plateaued at $K = 4$ and instead required $K = 5$ formats for stabilization (completion $\rightarrow$ ODQA $\rightarrow$ triplet $\rightarrow$ MC $\rightarrow$ TF). This shows that the iterative defense loop is order-sensitive: selecting formats by maximizing marginal utility enables a more sample-efficient plateau. However, once the plateau is reached, the final robustness level (measured by the max–min gap across formats) is comparable across both strategies. Thus, ordering primarily affects efficiency, how many formats are needed, but does not materially affect robustness once sufficiently informative formats are included.

**Generalization Patterns Across Models and Datasets.** To assess the broader applicability of our approach, we examine multi-format supervision across a range of models and datasets. We observe that the number of formats required for stabilization is model-dependent: while $K = 4$ is sufficient for most models on CounterFact-1k, it was not sufficient for Qwen3-0.6B, the smallest model we tested. This suggests that the degree of format-conditioned fragmentation depends on a model's inherent generalization capabilities. Despite this variability, our method remains consistently effective across architectures and scales: applying MEMIT$_{MF}$ with $K = 4$ reduces the max-min efficacy gap to below $0.05$ for all but two models (OLMo2-13B: $0.14$, Qwen3-0.6B: $0.53$). Full results are provided in Appendix C

We also evaluate dataset dependence by repeating iterative expansion on the PEP dataset with Llama-3.2-3B-Instruct. The procedure again converges at $K = 4$, though with a slightly different selection order (completion $\rightarrow$ TF $\rightarrow$ MC $\rightarrow$ triplet), illustrating that the informativeness of individual formats can vary across domains. Applying MEMIT$_{MF}$ with $K = 4$ achieves an efficacy of $0.90 \pm 0.07$ while single-format editing yields an efficacy of $0.38 \pm 0.18$. Together, these results indicate that while the specific format requirements may differ across models and datasets, the proposed multi-format supervision framework consistently enhances cross-format robustness.

## 4.2 AGENT-ASSISTED EXPANSION

While human-defined formats demonstrate the effectiveness of multi-format supervision, they require manual effort and cover only a limited design space. To scale beyond human-designed templates, we extend the adversarial defense loop with an agent-based generator that automates the attack phase. The agent proposes candidate formats, filters low-quality ones, and returns a set of viable formats. To ensure minimal format quality, we discard prompt templates whose pre-edit efficacy exceed 0.2, since these indicate that the model outputs the correct value even before editing and thus represent ill-formed or uninformative formats. By automating the generation process, the whole loop can be executed without human intervention.

Starting from $\mathcal{F}_{src} = \{f_{completion}\}$, we run the agent-assisted attack phase for 10 rounds, generating 10 candidate formats per round with GPT-5. Across these rounds the agent produced 67 valid formats, and the iterative expansion selected 6 formats, yielding a final defense set of 7 formats (completion + 6 discovered).

**Diversity of generated formats.** Although we attempted to guide the agent toward diverse formats through explicit instructions, quantifying diversity is intrinsically challenging. As a proxy for

semantic overlap, we compute pairwise semantic similarity using BERTScore (Zhang et al., 2020). The six human-designed formats exhibit an average similarity of 0.25. The full agent-generated pool of 67 candidates shows a higher similarity (0.33), suggesting limited diversity, but the six formats selected through iterative expansion achieve a reduced similarity of 0.29. This indicates that the selection process helps filter out redundant formats.

**Exposing vulnerable formats.** We next evaluate whether the agent-generated formats surface weaknesses that human-curated templates fail to capture. When edited with the six curated formats, the model achieves an average efficacy of $0.45 \pm 0.30$ on the six final agent-generated formats, with values ranging from 0.05 to 0.96. This indicates that agent-generated formats effectively expose vulnerable formats. For example, the `ClaimFalseThenCorrect` format (shown below) attains the lowest efficacy (0.05). This format explicitly embeds the original factual claim and requires a conditional correction, which stresses branching logic and reactivates pre-edit representations, which plain completion or simple TF/MC formats typically do not probe.

> **ClaimFalseThenCorrect format generated by the agent**
>
> {prompt} {target_true}
> If this claim is false in the updated records, output the corrected value; otherwise output {target_true}.
> Answer:

**Impact on generalization.** We also measure whether incorporating these formats improves cross-format generalization. Using the final defense set of 7 formats (completion + 6 agent-generated) for editing and evaluating on the five human-curated formats (excluding completion), the model achieves an efficacy of $0.67 \pm 0.34$, compared to $0.59 \pm 0.15$ when edited on completion alone. The model generalizes well on ODQA and triplet formats (1.00 and 0.99), but remains weak on True/False and Yes/No (0.31 and 0.32). This suggests that agent-assisted expansion broadens coverage and improves generalization to some formats, but still does not fully capture the full range of format variations, leaving room for further refinement.

Although the agent-assisted expansion suggests a path toward scalable discovery of vulnerable formats, enabling editing methods to proactively inoculate against failures, we see substantial room for improvement in generating diverse, high-quality formats. Understanding the extent of the long-tail distribution of possible formats and determining how to defend against it remain important open questions.

## 5 RELATED WORK

### 5.1 KNOWLEDGE EDITING

Knowledge editing methods can be broadly categorized into *parametric approaches*, which directly modify model parameters to encode new associations (Fang et al., 2025; Li et al., 2024; Hartvigsen et al., 2023; Meng et al., 2023; 2022; Mitchell et al., 2022a; Dai et al., 2022; De Cao et al., 2021; Zhu et al., 2020), and *non-parametric approaches*, which preserve parameters and instead alter predictions through external conditioning or retrieval (Cohen et al., 2024; Zeng et al., 2024; Zheng et al., 2023; Zhong et al., 2023; Mitchell et al., 2022b). Representative parametric methods include ROME, MEMIT and AlphaEdit, which update MLP projection weights to insert factual associations, while IKE serves as a representative non-parametric method that applies demonstration-style in-context updates. In this work, we focus on parametric editing, which is more directly tied to the model's representational structure.

Existing benchmarks cover a range of evaluation dimensions, including relation extraction (Levy et al., 2017), counterfactual editing (Meng et al., 2022), comprehensive evaluation (Zhang et al., 2024), multi-task evaluation (Zeng et al., 2024), multi-hop reasoning (Zhong et al., 2023), long-form generation (Rosati et al., 2024) and logical consistency (Cohen et al., 2024). Our work complements these by introducing a novel evaluation targeting *task-format generalization*, extending prior evaluations that focus on generalization across paraphrased inputs.

## 5.2 SEMANTIC CONSISTENCY

Semantic consistency concerns whether a model maintains invariant predictions when the input form is altered without changing its meaning (Jang et al., 2022; Elazar et al., 2021). Prior work has investigated this property under a variety of input perturbations, including *paraphrasing* (Jang et al., 2022; Elazar et al., 2021; Jin et al., 2020), *syntactic transformations* (Ravichander et al., 2020), *translation* (Wang et al., 2025; Qi et al., 2023), *contextual interference* (Xie et al., 2025; Yang et al., 2024) and *prompt formatting* (Kang et al., 2025; Sclar et al., 2024; Zhao et al., 2021). Our work focuses on the underexplored dimension of *task-format variation*, which goes beyond paraphrase or syntactic rewrites and captures shifts across heterogeneous tasks such as multiple-choice and true/false question answering.

Whereas prior studies primarily evaluate unedited models, we examine how semantic consistency is affected by *knowledge editing*. Specifically, we analyze whether instance-level edits generalize across task formats and how editing affects consistency.

## 5.3 KNOWLEDGE LOCALIZATION

Another line of research investigates where factual knowledge is stored within language models. Early work argue that transformer feed-forward layers operate as *key–value memories* (Geva et al., 2021), motivating attempts to localize factual associations to specific internal units. Building on this view, Dai et al. (2022) propose *knowledge neurons*, identifying neurons whose activations are causally linked to factual recall, while Meng et al. (2022) leverage *causal tracing* to identify target modules for editing. These approaches rest on a *localization assumption*: that individual facts can be pinpointed to a small set of neurons or parameters.

Recent findings have called this assumption into question. Chen et al. (2025b; 2024) introduce the notion of *degenerate knowledge neurons*, showing that distinct sets of neurons can encode the same fact and that such neurons also participate in storing other, different facts. Chen et al. (2025a) argue that knowledge localization is inherently query-dependent: the neurons activated for a fact vary across paraphrased inputs, suggesting a shift from fact localization to *query localization*.

Our work provides complementary evidence for this broader perspective. Whereas prior analyses highlight the query-dependence of localization, we demonstrate that edits occupy distinct *task-format-specific subspaces*. This finding supports a *distributed knowledge subspace hypothesis*, indicating that factual knowledge is redundantly and contextually encoded or accessed across multiple format-dependent subspaces, which in turn explains the cross-format generalization failures observed in our experiments.

## 6 CONCLUSION

We investigate whether parametric knowledge editing in large language models generalizes across task formats. The results show that edits often fail to transfer beyond the source format, and that even highly frequent facts do not necessarily yield a unified, format-invariant mechanism. Representation-level analyses further reveal that edit directions cluster by format, suggesting that knowledge is distributed across distinct, format-dependent representational subspaces for the same fact. We refer to this perspective as the *distributed knowledge subspace hypothesis*. This view underscores the need for approaches that explicitly identify and reconcile distributed representations of knowledge. While our work provides empirical evidence characterizing this phenomenon, theoretical formalization remains an important direction. We hope these findings help open this line of investigation and provide a foundation for more principled editing methods that achieve robust, format-invariant behavior.

## REPRODUCIBILITY STATEMENT

We have included all necessary experimental details in the main text and appendix, including dataset construction and evaluation protocols. Code and data will be released upon publication.

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

## A MULTI-FORMAT DATA CONSTRUCTION

Table 2: Multi-format templates used in our evaluation framework. Each factual edit $(s, r, o^c \Rightarrow o^*)$ is instantiated across these formats.

| Format | Template | Edit Targets |
|--------|----------|--------------|
| Completion | {completion_template$(s, r)$} | $o^c \Rightarrow o^*$ |
| Triplet | ( subject, relation, object )
{triplet_template$(s, r)$} | $o^c \Rightarrow o^*$ |
| ODQA | Question: {whqa_template$(s, r)$}
Answer: | $o^c \Rightarrow o^*$ |
| MC | Output only answer letter.
Question: {whqa_template$(s, r)$}
Options: A. {$o^*$ / $o^c$} B. {$o^c$ / $o^*$}
Answer: | {B $\Rightarrow$ A / A $\Rightarrow$ B} |
| TF | True or False?
Statement: {completion_template$(s, r)$} {$o^*$ / $o^c$}
Answer: | {False $\Rightarrow$ True / True $\Rightarrow$ False} |
| YN | Yes or No?
Statement: {ynqa_template$(s, r)$} {$o^*$ / $o^c$}?
Answer: | {No $\Rightarrow$ Yes / Yes $\Rightarrow$ No} |

Table 3: Multi-format examples for the target edit *(Eiffel Tower, location, Paris $\Rightarrow$ Rome)*.

| Format | Example Prompt | Edit Targets |
|--------|----------------|--------------|
| Completion | `Eiffel Tower is located in` | Paris $\Rightarrow$ Rome |
| Triplet | `( subject, relation, object )`
`( Eiffel Tower, location,` | Paris $\Rightarrow$ Rome |
| ODQA | `Question:  Where is Eiffel Tower located?`
`Answer:` | Paris $\Rightarrow$ Rome |
| MC | `Output only answer letter.`
`Question:  Where is Eiffel Tower located?`
`Options:  A. Rome B. Paris`
`Answer:` | B $\Rightarrow$ A |
| TF | `True or False?`
`Statement:  Eiffel Tower is located in Rome`
`Answer:` | False $\Rightarrow$ True |
| YN | `Yes or No?`
`Question:   Is Eiffel Tower located in Rome?`
`Answer:` | No $\Rightarrow$ Yes |

Table 4: Relational templates for `completion_template` and `triplet_template`.

| Relation | completion_template | triplet_template |
|---|---|---|
| P17 | $\{s\}$ is located in | ( $\{s\}$, country, |
| P19 | $\{s\}$ was born in | ( $\{s\}$, place of birth, |
| P20 | $\{s\}$ died in | ( $\{s\}$, place of death, |
| P27 | $\{s\}$ is a citizen of | ( $\{s\}$, country of citizenship, |
| P30 | $\{s\}$ is located in | ( $\{s\}$, continent, |
| P36 | The capital of $\{s\}$ is | ( $\{s\}$, capital, |
| P37 | The official language of $\{s\}$ is | ( $\{s\}$, official language, |
| P39 | $\{s\}$ has the position of | ( $\{s\}$, position held, |
| P101 | $\{s\}$ works in the field of | ( $\{s\}$, field of work, |
| P103 | The native language of $\{s\}$ is | ( $\{s\}$, native language, |
| P106 | The occupation of $\{s\}$ is | ( $\{s\}$, occupation, |
| P108 | $\{s\}$ works for | ( $\{s\}$, employer, |
| P127 | $\{s\}$ is owned by | ( $\{s\}$, owned by, |
| P131 | $\{s\}$ is located in | ( $\{s\}$, located in the administrative territorial entity, |
| P136 | The genre played by $\{s\}$ is | ( $\{s\}$, genre, |
| P138 | $\{s\}$ is named after | ( $\{s\}$, named after, |
| P140 | $\{s\}$ is affiliated with the religion of | ( $\{s\}$, religion, |
| P159 | The headquarter of $\{s\}$ is in | ( $\{s\}$, headquarters location, |
| P176 | $\{s\}$ is produced by | ( $\{s\}$, manufacturer, |
| P178 | $\{s\}$ is developed by | ( $\{s\}$, developer, |
| P190 | $\{s\}$ is a twin city of | ( $\{s\}$, twinned administrative body, |
| P264 | $\{s\}$ is represented by music label | ( $\{s\}$, record label, |
| P276 | $\{s\}$ is located in | ( $\{s\}$, location, |
| P364 | The original language of $\{s\}$ is | ( $\{s\}$, original language of film or TV show, |
| P407 | $\{s\}$ was written in | ( $\{s\}$, language of work or name, |
| P413 | $\{s\}$ plays in the position of | ( $\{s\}$, position played on team, |
| P449 | $\{s\}$ was originally aired on | ( $\{s\}$, original network, |
| P463 | $\{s\}$ is a member of | ( $\{s\}$, member of, |
| P495 | $\{s\}$ was created in | ( $\{s\}$, country of origin, |
| P740 | $\{s\}$ was founded in | ( $\{s\}$, location of formation, |
| P937 | $\{s\}$ used to work in | ( $\{s\}$, work location, |
| P1303 | $\{s\}$ plays the | ( $\{s\}$, instrument, |
| P1412 | $\{s\}$ used to communicate in | ( $\{s\}$, languages spoken, written or signed, |
| PEP | $\{s\}$ can $\{r\}$ | ( $\{s\}$, can $\{r\}$, |

Table 5: Relational templates for `whqa_template` and `ynqa_template`.

| Relation | whqa_template | ynqa_template |
|---|---|---|
| P17 | Where is $\{s\}$ located? | Is $\{s\}$ located in |
| P19 | Where was $\{s\}$ born? | Was $\{s\}$ born in |
| P20 | Where did $\{s\}$ die? | Did $\{s\}$ die in |
| P27 | What country is $\{s\}$ a citizen of? | Is $\{s\}$ a citizen of |
| P30 | Where is $\{s\}$ located? | Is $\{s\}$ located in |
| P36 | What is the capital of $\{s\}$? | Is the capital of $\{s\}$ |
| P37 | What is the official language of $\{s\}$? | Is the official language of $\{s\}$ |
| P39 | What position does $\{s\}$ hold? | Does $\{s\}$ have the position of |
| P101 | What is the field of work for $\{s\}$? | Does $\{s\}$ work in the field of |
| P103 | What is the native language of $\{s\}$? | Is the native language of $\{s\}$ |
| P106 | What is the occupation of $\{s\}$? | Is the occupation of $\{s\}$ |
| P108 | Who does $\{s\}$ work for? | Does $\{s\}$ work for |
| P127 | Who owns $\{s\}$? | Is $\{s\}$ owned by |
| P131 | Where is $\{s\}$ located? | Is $\{s\}$ located in |
| P136 | What genre does $\{s\}$ play? | Is the genre played by $\{s\}$ |
| P138 | Who or what is $\{s\}$ named after? | Is $\{s\}$ named after |
| P140 | What religion is $\{s\}$ affiliated with? | Is $\{s\}$ affiliated with the religion of |
| P159 | Where is the headquarter of $\{s\}$? | Is the headquarter of $\{s\}$ in |
| P176 | Who produced $\{s\}$? | Is $\{s\}$ produced by |
| P178 | Who developed $\{s\}$? | Is $\{s\}$ developed by |
| P190 | What city is twinned with $\{s\}$? | Is $\{s\}$ a twin city of |
| P264 | What music label represents $\{s\}$? | Is $\{s\}$ represented by music label |
| P276 | Where is $\{s\}$ located? | Is $\{s\}$ located in |
| P364 | What is the original language of $\{s\}$? | Is the original language of $\{s\}$ |
| P407 | What language was $\{s\}$ written in? | Was $\{s\}$ written in |
| P413 | What position does $\{s\}$ play in? | Does $\{s\}$ play in the position of |
| P449 | Which network originally aired $\{s\}$? | Was $\{s\}$ originally aired on |
| P463 | What is $\{s\}$ a member of? | Is $\{s\}$ a member of |
| P495 | Where was $\{s\}$ created? | Was $\{s\}$ created in |
| P740 | Where was $\{s\}$ founded? | Was $\{s\}$ founded in |
| P937 | Where did $\{s\}$ use to work? | Did $\{s\}$ use to work in |
| P1303 | What instrument does $\{s\}$ play? | Does $\{s\}$ play the |
| P1412 | What language does $\{s\}$ use to communicate? | Does $\{s\}$ use to communicate in |
| PEP | What can $\{s\}$ $\{r\}$? | Can $\{s\}$ $\{r\}$ |

# B IMPLEMENTATION DETAILS

## B.1 PARAMETRIC KNOWLEDGE EDITING

### B.1.1 MEMIT

Across all models used in our experiments, we set the covariance adjustment hyperparameter $\lambda = 15000$. Covariance statistics are estimated based on 100,000 instances from Wikitext. The target value vector $v^*$ is optimized for 25 gradient steps with a learning rate of $5e{-}1$. We clamp the $L_2$ norm of edit delta such that it does not exceed 0.75 times the norm of the original hidden state. We set the weight decay parameter to 0.5 and KL regularization factor to 0.0625. We restrict the number of edits to one, which is sufficient to reveal the generalization failure. Unless otherwise noted, we follow the original implementation of MEMIT (Meng et al., 2023).

On Llama-3.2-3B-Instruct, Llama-3.1-8B-Instruct, OLMo2-1B-Instruct and OLMo2-13B-Instruct, we set the target layers $\mathcal{R} = \{2, 3, 4, 5, 6, 7\}$. On Llama-3.2-1B-Instruct, we set the target layers $\mathcal{R} = \{0, 1, 2, 3, 4, 5\}$. On OLMo2-7B-Instruct, we set the target layers $\mathcal{R} = \{3, 4, 5, 6, 7, 8\}$. On Qwen3-0.6B-Instruct, we set the target layers $\mathcal{R} = \{1, 2, 3, 4, 5, 7\}$. On Qwen3-1.7B-Instruct, we set the target layers $\mathcal{R} = \{3, 4, 5, 6, 7, 8\}$. On Qwen3-4B-Instruct, we set the target layers $\mathcal{R} = \{6, 7, 8, 9, 10, 11\}$. On Qwen3-8B-Instruct, we set the target layers $\mathcal{R} = \{5, 6, 7, 8, 9, 10\}$. On Qwen3-14B-Instruct, we set the target layers $\mathcal{R} = \{7, 8, 9, 10, 11, 12\}$.

### B.1.2 ROME

We use the final layer of the MEMIT's target layers as the target layer. The other hyperparameters follow the MEMIT.

### B.1.3 ALPHAEDIT

We set the nullspace threshold to 2e-2 and L2 factor to 10. The other hyperparameters follow the MEMIT.

## B.2 IN-CONTEXT KNOWLEDGE EDITING

We adopt a few-shot prompting scheme for in-context knowledge editing (IKE) (Zheng et al., 2023), where the model is explicitly instructed to remain consistent with the injected knowledge. The prompt template consists of three main components: the *instruction*, which directs the model to always align its answers with the provided knowledge triplets; the *demonstrations*, which include a small set of updated knowledge examples presented both as triplets and as corresponding natural language statements; and the *final query*, which is the actual target edit to be tested and is appended at the end of the template. Note that the demonstrations contain both *knowledge update* and *knowledge retain* cases.

The prompt example shows a concrete instantiation of this template. Several demonstration triplets are included (e.g., *Subair, occupation, composer*), each followed by a natural language realization of the updated knowledge. The final query then presents a new triplet (*Eiffel Tower, location, Rome*), prompting the model to generate an answer consistent with this modification.

---

**Prompt Example**

You are given updated knowledge in the form of (subject, relation, object). Always answer queries consistently with this updated knowledge.

—

Updated knowledge: (Subair, occupation, composer)
The occupation of Subair is composer.
—
Updated knowledge: (Jean-Pierre Dionnet, languages spoken, written or signed, Spanish)
Georges Pompidou used to communicate in French.
—
Updated knowledge: (Umayyad Caliphate, capital, Athens)
The capital of Umayyad Caliphate is Athens.
—
Updated knowledge: (Georges Bernier, native language, Russian)
The native language of Jean-Luc Picard is French.
—
Updated knowledge: (Bong Jung-keun, position played on team, outfielder)
Bong Jung-keun plays in the position of outfielder.
—

Now answer the following query based on the updated knowledge:

Updated knowledge: (Eiffel Tower, location, Rome)
Eiffel Tower is located in

---

## B.3 AGENT-ASSISTED EXPANSION

For format generation, we employed GPT-5. We filtered out invalid formats that triggered preprocessing errors, as well as those with pre-edit efficacy greater than 0.2, since high performance prior to editing indicates that the format itself may be problematic.

---

**Summary of Prompt Design for Automated Format Generation**

**Role**    Design diagnostically robust formats to test whether a knowledge-edited model has internalized a new fact consistently across prompts.

**Knowledge-Edit Semantics**
- {target_new}: **The New Truth** (treated as TRUE).
- {target_true}: **The Outdated Fact** (treated as FALSE).

**Allowed Placeholders (only these four)**
- {subject}: the entity (e.g., "Danielle Darrieux").
- {prompt}: base template containing a single {} slot for {subject}.
- {target_new}, {target_true}.

**Constraint:** {prompt} may be preceded only by optional whitespace; no explicit newline (\n) before it.

**Mandate 1: Consistency**    All formats must align with a world where {target_new} is TRUE and {target_true} is FALSE; avoid prompts whose internal logic would require denying the new truth to reach the correct answer.

**Mandate 2: Structural Diversity**    Cover varied modalities to reveal blind spots:
- Multiple Choice (fixed options).
- True/False.
- Open-Ended generation (direct token production).

**Design Principles**
- Keep internal logic self-consistent (no inverted conditions).
- In MCQ, ensure the correct key maps to {target_new} unambiguously.
- Structured outputs allowed (e.g., JSON). To emit literal braces when templating, use double braces: {{...}}.

**Output Requirement**    Return a **single, valid JSON object** per run; use only the four placeholders above; do not invent new placeholders.

**Authoring Task (for generation runs)**
1. Review given existing_formats to map covered "cognitive territory".
2. Propose 10 new formats that are logically sound, adhere to the Knowledge-Edit semantics, and probe uncovered areas.

---

# C ADDITIONAL RESULTS ON CROSS-FORMAT GENERALIZATION

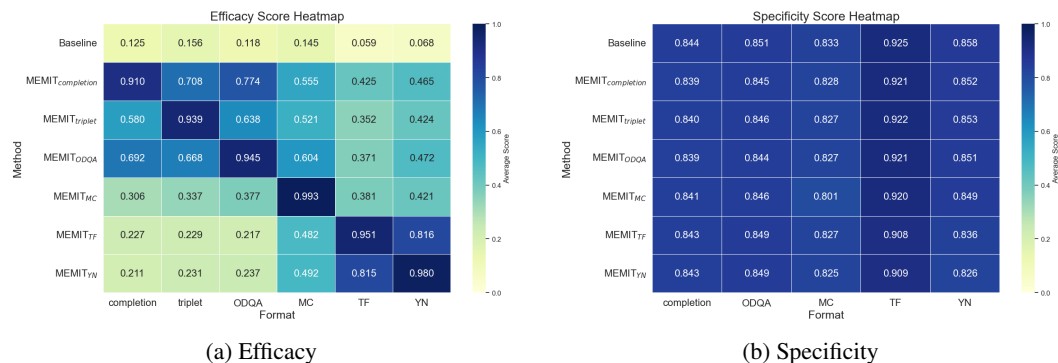

(a) Efficacy        (b) Specificity

Figure 6: Cross-format generalization results on Llama-3.2-3B-Instruct and CounterFact-1k when edits are performed in single source formats. Each cell reports the average score across all samples for each format.

Table 6: Cross-format generalization results on CounterFact-1k and MEMIT.

| Model | Mean $\pm$ Std | Max | Min | Max - Min |
|---|---|---|---|---|
| Llama-3.2-1B-Instruct | $0.76 \pm 0.15$ | 0.94 | 0.48 | 0.46 |
| + MEMIT$_{MF}$ ($K = 4$) | $0.97 \pm 0.02$ | 1.00 | 0.95 | 0.05 |
| Llama-3.2-3B-Instruct | $0.64 \pm 0.19$ | 0.91 | 0.43 | 0.48 |
| + MEMIT$_{MF}$ ($K = 4$) | $0.99 \pm 0.01$ | 1.00 | 0.97 | 0.03 |
| Llama-3.1-8B-Instruct | $0.86 \pm 0.10$ | 1.00 | 0.75 | 0.25 |
| + MEMIT$_{MF}$ ($K = 4$) | $0.99 \pm 0.02$ | 1.00 | 0.96 | 0.04 |
| OLMo2-1B-Instruct | $0.63 \pm 0.37$ | 0.99 | 0.09 | 0.90 |
| + MEMIT$_{MF}$ ($K = 4$) | $0.99 \pm 0.01$ | 1.00 | 0.98 | 0.02 |
| OLMo2-7B-Instruct | $0.81 \pm 0.16$ | 1.00 | 0.62 | 0.38 |
| + MEMIT$_{MF}$ ($K = 4$) | $1.00 \pm 0.00$ | 1.00 | 0.99 | 0.01 |
| OLMo2-13B-Instruct | $0.58 \pm 0.14$ | 0.77 | 0.43 | 0.34 |
| + MEMIT$_{MF}$ ($K = 4$) | $0.88 \pm 0.05$ | 0.94 | 0.80 | 0.14 |
| Qwen3-0.6B | $0.47 \pm 0.29$ | 0.95 | 0.03 | 0.92 |
| + MEMIT$_{MF}$ ($K = 4$) | $0.77 \pm 0.19$ | 0.99 | 0.46 | 0.53 |
| Qwen3-1.7B | $0.82 \pm 0.10$ | 0.96 | 0.69 | 0.27 |
| + MEMIT$_{MF}$ ($K = 4$) | $0.99 \pm 0.02$ | 1.00 | 0.96 | 0.04 |
| Qwen3-4B | $0.74 \pm 0.19$ | 0.97 | 0.51 | 0.46 |
| + MEMIT$_{MF}$ ($K = 4$) | $0.99 \pm 0.01$ | 1.00 | 0.97 | 0.03 |
| Qwen3-8B | $0.82 \pm 0.15$ | 0.99 | 0.63 | 0.36 |
| + MEMIT$_{MF}$ ($K = 4$) | $0.99 \pm 0.01$ | 1.00 | 0.98 | 0.02 |
| Qwen3-14B | $0.88 \pm 0.13$ | 1.00 | 0.68 | 0.32 |
| + MEMIT$_{MF}$ ($K = 4$) | $1.00 \pm 0.00$ | 1.00 | 0.99 | 0.01 |

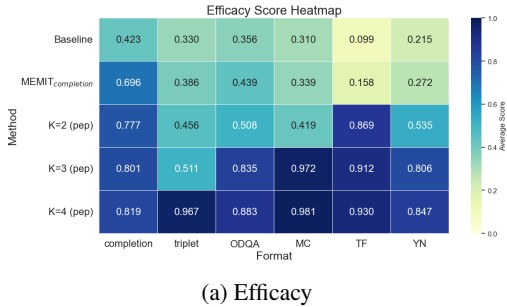

(a) Efficacy

Figure 7: Cross-format generalization results under multi-format supervision with iterative expansion on Llama-3.2-3B-Instruct and PEP with MEMIT. The supervision set (source formats) is progressively expanded (completion $\rightarrow$ TF $\rightarrow$ MC $\rightarrow$ triple), and terminates at $K = 4$ when efficacy scores on the other formats exceed the completion.

# D DISTRIBUTION OF EDIT DIRECTIONS

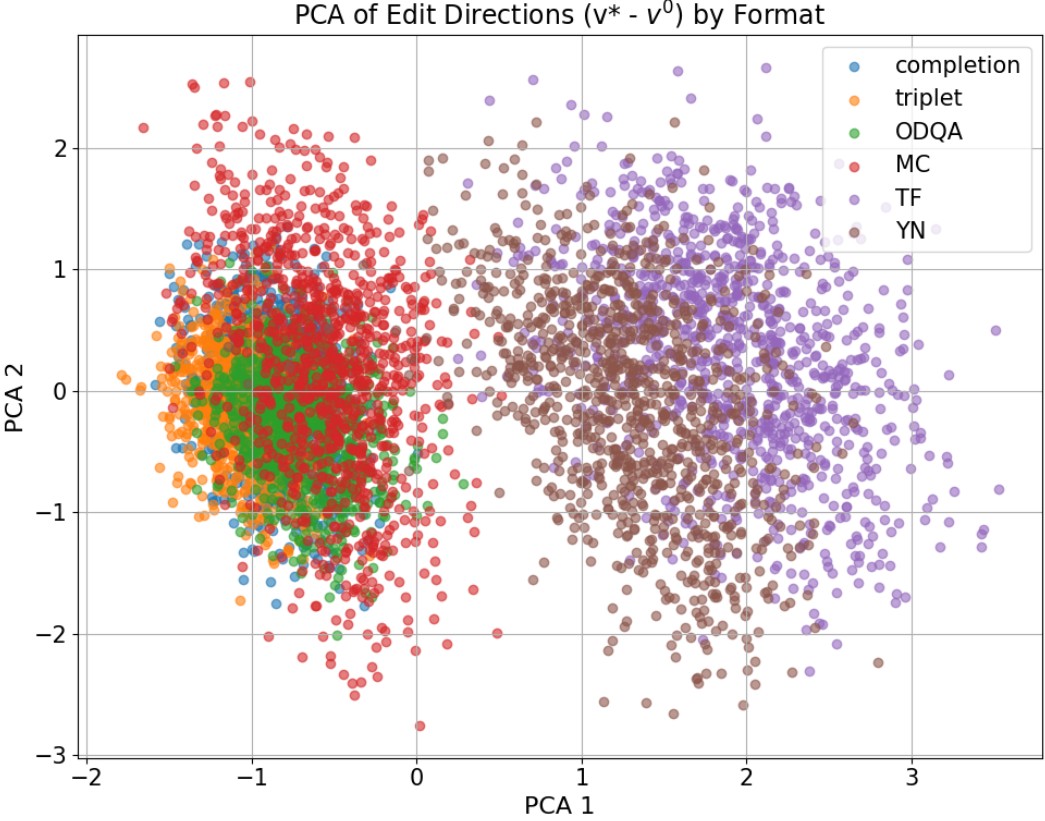

Figure 8: PCA visualization of final edit directions $(v^* - v^0)$ of Llama-3.2-3B-Instruct across formats on CounterFact-1k. Each point corresponds to the optimized edit direction for a given triple in one format $f$. The visible clustering by format indicates that different input styles drive editing into distinct subspaces, reinforcing the observation that edited knowledge does not converge to a shared representation across formats.

# E    CAUSAL TRACING

Figure 9: Causal tracing with Llama-3.2-3B-Instruct for different formats.

# F  IDENTIFYING CRITICAL MLP LAYERS

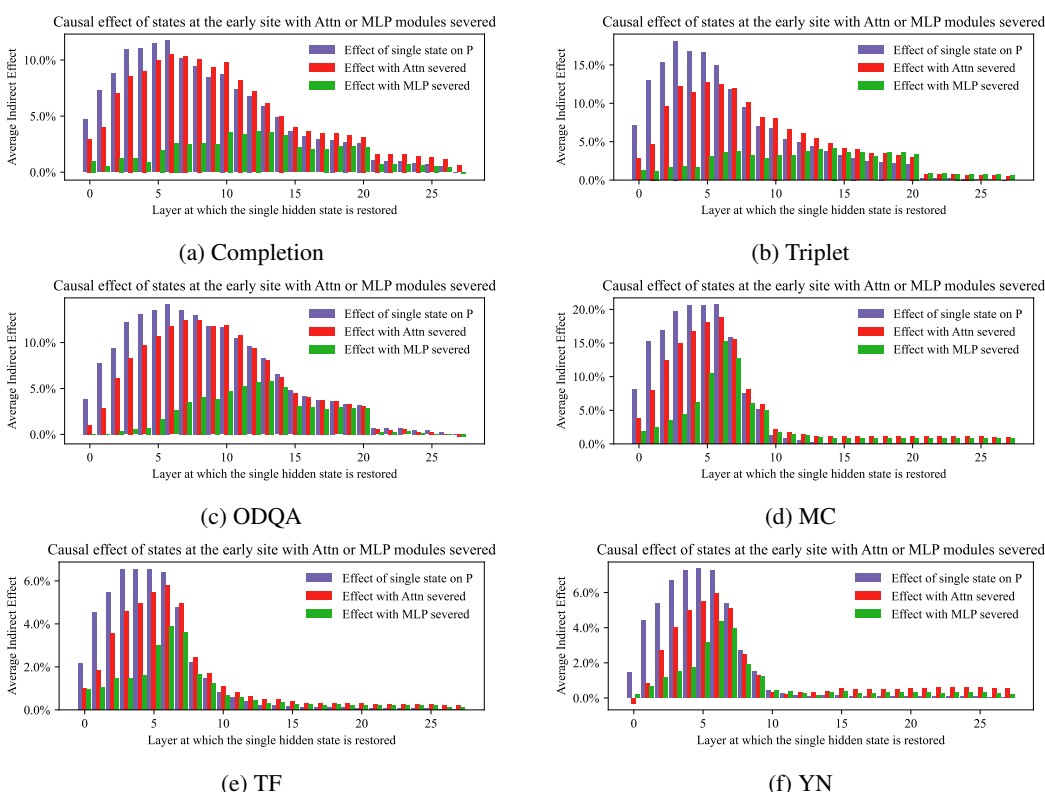

Figure 10: Critical MLP layers of Llama-3.2-3B-Instruct for different formats.

# G  THE USE OF LARGE LANGUAGE MODELS

We used large language models as assistive tools for writing, Specifically, they were employed to check grammar, refine wording and flag potential inconsistencies. All final texts were carefully reviewed and validated by the authors.

