# OpenReview forum: "Distributed Knowledge Storage Hypothesis: Evidence from Generalization Failure in Knowledge Editing"
_ICLR.cc/2026/Conference — Submitted to ICLR 2026_

### Official Review · Reviewer_1i3Q · 2025-10-29

**Soundness:** 3
**Presentation:** 3
**Contribution:** 3
**Rating:** 4
**Confidence:** 4

**Summary:**

This paper examines cross-format generalization in LLM knowledge editing. It finds that edits made through methods like MEMIT and IKE often fail to transfer across different task formats. This result supports the distributed knowledge storage hypothesis which states that facts are encoded in format-specific subspaces. To solve this issue the study proposes a multi-format supervised iterative approach to improve cross-format efficacy.

**Strengths:**

**Strength 1**

Comprehensive cross-format evaluation for LLM knowledge editing, covering six diverse task formats to capture real-world usage scenarios.

**Strength 2**

Provides robust empirical and representation-level evidence (e.g., format-clustered edit directions) that substantiates the distributed knowledge storage hypothesis.

**Strength 3**

Proposes a practical multi-format supervised iterative method that effectively improves cross-format edit robustness while maintaining specificity.

**Weaknesses:**

**Weakness 1. Experimental Scope:**

The experiments are restricted to a single open-source LLM (Llama-3.2-3B-Instruct) and a single benchmark dataset (CounterFact-1k). It remains unclear whether the observed phenomena hold for larger, more diverse models (e.g., DeepSeek, Qwen) or other factual domains. Broader validation would strengthen generality claims.

**Weakness 2. Theoretical Underdevelopment of Core Hypothesis:**

The distributed knowledge storage hypothesis relies heavily on empirical observations (e.g., format-clustered edit directions) but lacks rigorous theoretical formalization.

**Weakness 3. Incomplete Baseline and Related Work Coverage:**

Only two editing methods (MEMIT, IKE) are systematically evaluated, excluding recent relevant approaches (e.g., AlphaEdit, PMET) with advanced localization.

**Questions:**

I will happily raise my score if my concerns are addressed.

**Question 1**

Do the observed generalization failures persist in larger models (e.g., Llama-3-70B, GPT-type models) or for multilingual/fine-tuned LLMs? Are there any preliminary results or hypotheses on scaling effects?

**Question 2**

Robustness to Task Domain: Does the fragmentation of edits and the distributed subspace hypothesis hold for other factual relations (e.g., temporal, causal, commonsense) or less-structured domains (e.g., multi-hop facts, subjective knowledge)?

**Question 3**

Baseline Selection: Why were methods like AlphaEdit, PMET, or ROME variants omitted from benchmarking experiments? Would incorporating these alter the conclusions about universality of generalization failure?

---

> ### Author Response · Authors · 2025-11-18
>
> Thank you for the helpful feedback.
>
> ## Extending experimental scope (Models, Datasets, Editing Methods)
> **We have incorporated the additional experimental results into the revised manuscript, specifically in Sec. 3.2, 4.1 and Appendix C.**
>
> **Models.** To assess the generality of our findings, we extended our evaluation to a broad set of models spanning three model families (Llama, OLMo2, Qwen3) and parameter ranges from 0.6B to 14B. Tested on $Counterfact_{1k}$ with $MEMIT_{completion}$, we observe that larger models have smaller cross-format variance in general, but the largest models we tested (13B-14B sized) still show a sizeable gap. Our analysis is limited to models up to 14B parameters due to computational constraints, and exploring whether clearer scaling trends emerge at larger scales is an important direction to be investigated. Nevertheless, **we consistently observe cross-format divergence across all scales and model families tested, confirming that the phenomenon is not specific to a single model family or size.** Notably, when applying our proposed multi-format supervision ($MEMIT_{MF}$, K = 4), the Max - Min gap collapses to below 0.05 for all models except two (OLMo2-13B: 0.14, Qwen3-0.6B: 0.53), demonstrating that multi-format alignment effectively mitigates the format-conditioned fragmentation for diverse models.
>
> ### Llama Family
> | Model           | Mean ± Std        | Min    | Max    | Max - Min |
> |-----------------|--------------------|--------|--------|---------|
> | Llama-3.2-1B    | 0.76 ± 0.15    | 0.48  | 0.94  | 0.46   |
> | Llama-3.2-3B    | 0.64 ± 0.19    | 0.43  | 0.91  | 0.48   |
> | Llama-3.1-8B    | 0.86 ± 0.10    | 0.75  | 1.00  | 0.25   |
>
> ---
>
> ### OLMo2 Family
> | Model           | Mean ± Std        | Min    | Max    | Max - Min |
> |-----------------|--------------------|--------|--------|---------|
> | OLMo2-1B        | 0.63 ± 0.37    | 0.09  | 0.99  | 0.90   |
> | OLMo2-7B        | 0.81 ± 0.16    | 0.62  | 1.00  | 0.38   |
> | OLMo2-13B       | 0.58 ± 0.14    | 0.43  | 0.77  | 0.34   |
>
> ---
>
> ### Qwen3 Family
> | Model           | Mean ± Std        | Min    | Max    | Max - Min |
> |-----------------|--------------------|--------|--------|---------|
> | Qwen3-0.6B      | 0.47 ± 0.29    | 0.03  | 0.95  | 0.92   |
> | Qwen3-1.7B      | 0.82 ± 0.10    | 0.69  | 0.96  | 0.27   |
> | Qwen3-4B        | 0.74 ± 0.19    | 0.51  | 0.97  | 0.46   |
> | Qwen3-8B        | 0.82 ± 0.15    | 0.63  | 0.99  | 0.36   |
> | Qwen3-14B       | 0.88 ± 0.13    | 0.68  | 1.00  | 0.32   |
>
> **Datasets.** To examine whether our findings generalize to other types of relations, we extended our experiments to the PEP (Physical Event Plausibility) dataset, which tests commonsense physical reasoning [1]. Using our standard setting (Llama-3.2-3B-Instruct + MEMIT), we found that the same pattern of cross-format fragmentation persists. Editing only with the completion format yields an average efficacy of 0.38 +- 0.18 (min = 0.16, max = 0.70), whereas applying our multi-format supervision (K = 4) substantially improves performance to 0.90 +- 0.07 (min = 0.82, max = 0.98). These results confirm that the **format-conditioned behavior observed in our main experiments also arises in commonsense relational editing**, further reinforcing the generality of the underlying phenomenon.
>
> **Editing Methods.** To ensure that our findings are not tied to a particular method, we additionally evaluated two parametric editing approaches: ROME and AlphaEdit. Tested on $Counterfact_{1k}$ and Llama-3.2-3B-Instruct, the efficacies when edited only with the completion format were: MEMIT (0.64 +- 0.19, min = 0.43, max = 0.91), ROME (0.79 +- 0.22, min = 0.54, max = 1.00) and AlphaEdit (0.70 +- 0.21, min = 0.47, max = 0.97). While the specific performance levels vary across methods, they all exhibit large gaps (> 0.45) between the min and max efficacies. **This confirms that the cross-format divergence we study is prevalent across editing methods.**
>
> [1] Ian Porada, Kaheer Suleman, Adam Trischler, and Jackie Chi Kit Cheung. 2021. Modeling Event Plausibility with Consistent Conceptual Abstraction. In Proceedings of the 2021 Conference of the North American Chapter of the Association for Computational Linguistics: Human Language Technologies, pages 1732–1743, Online. Association for Computational Linguistics.
>
> ## Theoretical formalization
>
> We agree that theoretical formalization is indeed an important direction, and we hope our empirical findings help open up this line of investigation. We have incorporated this into the revised version.

---

> ### Author Response · Authors · 2025-11-26
> **Have our clarifications addressed the concerns?**
>
> Dear Reviewer 1i3Q,
>
> Thank you for your constructive feedback. We have made revisions in response to your comments to clarify and strengthen the contribution of our work. We welcome any additional suggestions.
>
> Please let us know whether the revisions address your concerns.
>
> Thank you again for your time and consideration.

---

> > ### Comment · Reviewer_1i3Q · 2025-11-26
> >
> > Thank you to the authors for the clear and thorough responses. All of my concerns have been addressed in the rebuttal. I appreciate the effort and care put into the replies.

---

### Official Review · Reviewer_U1ok · 2025-10-30

**Soundness:** 2
**Presentation:** 2
**Contribution:** 2
**Rating:** 2
**Confidence:** 3

**Summary:**

This paper investigates the limitations of current knowledge editing methods, showing that edits applied in one format cannot be effectively generalized to other formats. The authors conduct experiments across six task formats, evaluating the cross-format generalization performance of knowledge editing through various metrics, and analyzing the distribution of knowledge across different formats in the representation space. They propose the distributed knowledge storage hypothesis, suggesting that factual knowledge in LLMs is encoded within multiple format-specific subspaces. Finally, the paper introduces an adversarial procedure called Multi-Format Supervision with Iterative Expansion, which enhances cross-format generalization performance.

**Strengths:**

1. The authors design six distinct task formats to explore the generalization capability of knowledge editing across different forms. The data construction process is relatively comprehensive, and the experimental analysis is persuasive.
2. The analysis of representation distributions through PCA projection intuitively demonstrates the clear separation of edit directions across different task formats, supporting the distributed knowledge storage hypothesis.
3. The Multi-Format Supervision with Iterative Expansion method is simple and well-motivated. By using an agent to assist in generating new task formats, it improves cross-format generalization with minimal human intervention.

**Weaknesses:**

1. All experiments are conducted only on the Llama-3.2-3B-Instruct model and the CounterFact-1k dataset, without validation on other models or alternative knowledge-editing datasets. This limits the generality of the conclusions, including both the distributed knowledge storage hypothesis and the proposed method.
2. Only MEMIT and IKE are evaluated as knowledge editing baselines, lacking comparison with stronger or more diverse editing approaches.
3. The analysis of agent-assisted task format generation is relatively shallow, with no quantitative evaluation of the generated formats’ quality, diversity, or their actual impact on improving generalization.

**Questions:**

1. Beyond the Llama-3.2-3B-Instruct and CounterFact-1k setup, have you observed similar cross-format generalization failures on other state-of-the-art models or larger-scale editing datasets?
2. Is there a principled reason multi-format supervision plateaued at $K=4$ formats (as in Figure 5), or is this dataset/model dependent? Does the order of format addition in the iterative defense loop affect robustness or efficiency?
3. Could you provide more details about the agent-assisted experimental results?

---

> ### Author Response · Authors · 2025-11-18
>
> Thank you for the valuable feedback.
>
> ## Extending experimental scope (Models, Datasets, Editing Methods)
> **We have incorporated the additional experimental results into the revised manuscript, specifically in Sec. 3.2, 4.1 and Appendix C.**
>
> **Models.** To assess the generality of our findings, we extended our evaluation to a broad set of models spanning three model families (Llama, OLMo2, Qwen3) and parameter ranges from 0.6B to 14B. Tested on $Counterfact_{1k}$ with $MEMIT_{completion}$, we observe that larger models have smaller cross-format variance in general, but the largest models we tested (13B-14B sized) still show a sizeable gap. Our analysis is limited to models up to 14B parameters due to computational constraints, and exploring whether clearer scaling trends emerge at larger scales is an important direction to be investigated. Nevertheless, **we consistently observe cross-format divergence across all scales and model families tested, confirming that the phenomenon is not specific to a single model family or size.** Notably, when applying our proposed multi-format supervision ($MEMIT_{MF}$, K = 4), the Max - Min gap collapses to below 0.05 for all models except two (OLMo2-13B: 0.14, Qwen3-0.6B: 0.53), demonstrating that multi-format alignment effectively mitigates the format-conditioned fragmentation for diverse models.
>
> ### Llama Family
> | Model           | Mean ± Std        | Min    | Max    | Max - Min |
> |-----------------|--------------------|--------|--------|---------|
> | Llama-3.2-1B    | 0.76 ± 0.15    | 0.48  | 0.94  | 0.46   |
> | Llama-3.2-3B    | 0.64 ± 0.19    | 0.43  | 0.91  | 0.48   |
> | Llama-3.1-8B    | 0.86 ± 0.10    | 0.75  | 1.00  | 0.25   |
>
> ---
>
> ### OLMo2 Family
> | Model           | Mean ± Std        | Min    | Max    | Max - Min |
> |-----------------|--------------------|--------|--------|---------|
> | OLMo2-1B        | 0.63 ± 0.37    | 0.09  | 0.99  | 0.90   |
> | OLMo2-7B        | 0.81 ± 0.16    | 0.62  | 1.00  | 0.38   |
> | OLMo2-13B       | 0.58 ± 0.14    | 0.43  | 0.77  | 0.34   |
>
> ---
>
> ### Qwen3 Family
> | Model           | Mean ± Std        | Min    | Max    | Max - Min |
> |-----------------|--------------------|--------|--------|---------|
> | Qwen3-0.6B      | 0.47 ± 0.29    | 0.03  | 0.95  | 0.92   |
> | Qwen3-1.7B      | 0.82 ± 0.10    | 0.69  | 0.96  | 0.27   |
> | Qwen3-4B        | 0.74 ± 0.19    | 0.51  | 0.97  | 0.46   |
> | Qwen3-8B        | 0.82 ± 0.15    | 0.63  | 0.99  | 0.36   |
> | Qwen3-14B       | 0.88 ± 0.13    | 0.68  | 1.00  | 0.32   |
>
> **Datasets.** To examine whether our findings generalize to other types of relations, we extended our experiments to the PEP (Physical Event Plausibility) dataset, which tests commonsense physical reasoning [1]. Using our standard setting (Llama-3.2-3B-Instruct + MEMIT), we found that the same pattern of cross-format fragmentation persists. Editing only with the completion format yields an average efficacy of 0.38 +- 0.18 (min = 0.16, max = 0.70), whereas applying our multi-format supervision (K = 4) substantially improves performance to 0.90 +- 0.07 (min = 0.82, max = 0.98). These results confirm that the **format-conditioned behavior observed in our main experiments also arises in commonsense relational editing**, further reinforcing the generality of the underlying phenomenon.
>
> **Editing Methods.** To ensure that our findings are not tied to a particular method, we additionally evaluated two parametric editing approaches: ROME and AlphaEdit. Tested on $Counterfact_{1k}$ and Llama-3.2-3B-Instruct, the efficacies when edited only with the completion format were: MEMIT (0.64 +- 0.19, min = 0.43, max = 0.91), ROME (0.79 +- 0.22, min = 0.54, max = 1.00) and AlphaEdit (0.70 +- 0.21, min = 0.47, max = 0.97). While the specific performance levels vary across methods, they all exhibit large gaps (> 0.45) between the min and max efficacies. **This confirms that the cross-format divergence we study is prevalent across editing methods.**
>
> [1] Ian Porada, Kaheer Suleman, Adam Trischler, and Jackie Chi Kit Cheung. 2021. Modeling Event Plausibility with Consistent Conceptual Abstraction. In Proceedings of the 2021 Conference of the North American Chapter of the Association for Computational Linguistics: Human Language Technologies, pages 1732–1743, Online. Association for Computational Linguistics.

---

> ### Author Response · Authors · 2025-11-18
>
> ## Details about the agent-assisted experimental results
>
> **We have reorganized Sec. 4.2 in the revised manuscript to incorporate the details shown below.**
>
> To ensure minimal format quality, we discarded prompt templates that triggered code-execution errors, and cases where pre-edit efficacy exceeded 0.2 (which indicates that the model already outputs the correct value even before editing, suggesting an ill-formed or uninformative format). After filtering, 67 out of 100 generated formats remained.
>
> Although we attempted to guide the agent toward diverse formats through explicit instructions, quantifying the diversity of task formats is intrinsically challenging. We attempted to quantify how semantically distinct the task formats are by measuring pairwise similarity using BERTScore [2] as a proxy for semantic overlap. For our human-designed formats, the average pairwise similarity across 6 formats was 0.25. The agent-generated pool of 67 candidates exhibited a higher similarity (0.33). After iterative expansion (resulting in 6 final agent-selected formats), the similarity decreased to 0.29, showing that the iterative selection process helped filter out redundant formats.
>
> To measure their actual impact on improving generalization, we edited the model using the final 7 formats (completion + 6 agent-generated ones) and evaluated generalization on the 5 human-curated formats (excluding completion). The resulting efficacy was 0.67 +- 0.34, while the baseline efficacy (edited with completion only) is 0.59 +- 0.15. The model generalized well on ODQA and triplet formats (1.00, 0.99), but remained weak on True/False and Yes/No (0.31, 0.32). This suggests that agent-assisted expansion broadens coverage and improves generalization to some formats, but still does not fully capture the full range of format variations, leaving room for improvement. We acknowledge that our agent-assisted expansion is an initial exploration of this direction.
>
> [2] Tianyi Zhang, Varsha Kishore, Felix Wu, Kilian Q. Weinberger, and Yoav Artzi. 2020. BERTScore: Evaluating Text Generation with BERT. In Proceedings of the International Conference on Learning Representations (ICLR).
>
> ## Is there a principled reason multi-format supervision plateaued at (K=4) formats (as in Figure 5), or is this dataset/model dependent?
>
> Our additional experiments indicate that this is both model-dependent and dataset-dependent. For instance, while K = 4 was sufficient for most models in our main experiments, it was not sufficient for Qwen3-0.6B, the smallest model we tested, suggesting that **the required number of formats depends on a model's inherent generalization ability.**
>
> To examine dataset dependence, we repeated the multi-format supervision with iterative expansion on the PEP dataset with Llama-3.2-3B-Instruct. The selected formats were completion => TF => MC => triplet, and performance plateaued again at K = 4, the same size as Counterfact, although the selection order differed slightly (Counterfact selects completion => TF => triplet => MC). While the optimal K was the same, the results suggest that the relative informativeness of specific formats can vary depending on the underlying data distribution.
>
> **We have included these analyses in Sec. 4.1 in the revised version.**
>
> ## Does the order of format addition in the iterative defense loop affect robustness or efficiency?
>
> We tested whether the order of format addition affects robustness or efficiency. Instead of selecting the lowest-scoring format among remaining candidates, we reversed the selection rule and chose the highest-scoring remaining format at each step. Under this alternative procedure, the model did not plateau at K = 4 and instead required K = 5 formats for stabilization (completion => ODQA => triplet => MC => TF). This shows that the iterative defense loop is order-sensitive, and that selecting formats based on maximizing marginal utility (our original design) leads to a more sample-efficient plateau. However, once the plateau was reached, the final robustness level was comparable to that of our original selection strategy. This suggests that **order affects efficiency (how many formats are needed), but does not materially affect robustness once sufficient formats have been included.**
>
> **We have included these results in Sec. 4.1 in the revision.**

---

> ### Author Response · Authors · 2025-11-26
> **Have our clarifications addressed the concerns?**
>
> Dear Reviewer U1ok,
>
> Thank you for your constructive feedback. We have made revisions in response to your comments to clarify and strengthen the contribution of our work. We welcome any additional suggestions.
>
> Please let us know whether the revisions address your concerns.
>
> Thank you again for your time and consideration.

---

### Official Review · Reviewer_ezRL · 2025-10-30

**Soundness:** 2
**Presentation:** 2
**Contribution:** 2
**Rating:** 4
**Confidence:** 4

**Summary:**

This paper investigates the problem of cross-format generalization of knowledge editing. The authors show that an edit that succeeds in the source format often fails in other formats and may even break previously consistent answers. From both behavioral and representation analysis (edited value vectors cluster by format), they argue that LLMs store factual knowledge in a distributed, format-dependent way and propose an iterative multi-format supervision strategy to harden edits against such failures.

**Strengths:**

* The problem is well-motivated and addresses a missing dimension in current benchmarks.
* The analysis is sound.
* The proposed agent-based system is simple and actionable.

**Weaknesses:**

* Limited experimental scope. All results are reported on a single CounterFact-1k subset and one 3B instruct model, which significantly reduces confidence in the generality of the findings.
* The contribution of the “distributed storage” concept is somewhat limited. Under the MEMIT framework, different formats naturally yield different keys, so edits are expected to occur in different locations. Under the IKE framework, the observed failures seem to stem more from the model’s format-sensitive reasoning/common-sense abilities—an area where a 3B model is clearly weak—rather than from how knowledge is actually stored.

**Questions:**

* What if we apply six edits simultaneously, one per format?
* Alternatively, could we collect a sufficiently large set of prompt prefixes for a subject, cluster them to identify the major key variants, and apply edits to all of them?
* Would such a multi-key / multi-format editing strategy address the observed issue?

---

> ### Author Response · Authors · 2025-11-18
>
> Thank you for the constructive reviews.
>
> ## Extending experimental scope (Models, Datasets, Editing Methods)
> **We have incorporated the additional experimental results into the revised manuscript, specifically in Sec. 3.2, 4.1 and Appendix C.**
>
> **Models.** To assess the generality of our findings, we extended our evaluation to a broad set of models spanning three model families (Llama, OLMo2, Qwen3) and parameter ranges from 0.6B to 14B. Tested on $Counterfact_{1k}$ with $MEMIT_{completion}$, we observe that larger models have smaller cross-format variance in general, but the largest models we tested (13B-14B sized) still show a sizeable gap. Our analysis is limited to models up to 14B parameters due to computational constraints, and exploring whether clearer scaling trends emerge at larger scales is an important direction to be investigated. Nevertheless, **we consistently observe cross-format divergence across all scales and model families tested, confirming that the phenomenon is not specific to a single model family or size.** Notably, when applying our proposed multi-format supervision ($MEMIT_{MF}$, K = 4), the Max - Min gap collapses to below 0.05 for all models except two (OLMo2-13B: 0.14, Qwen3-0.6B: 0.53), demonstrating that multi-format alignment effectively mitigates the format-conditioned fragmentation for diverse models.
>
> ### Llama Family
> | Model           | Mean ± Std        | Min    | Max    | Max - Min |
> |-----------------|--------------------|--------|--------|---------|
> | Llama-3.2-1B    | 0.76 ± 0.15    | 0.48  | 0.94  | 0.46   |
> | Llama-3.2-3B    | 0.64 ± 0.19    | 0.43  | 0.91  | 0.48   |
> | Llama-3.1-8B    | 0.86 ± 0.10    | 0.75  | 1.00  | 0.25   |
>
> ---
>
> ### OLMo2 Family
> | Model           | Mean ± Std        | Min    | Max    | Max - Min |
> |-----------------|--------------------|--------|--------|---------|
> | OLMo2-1B        | 0.63 ± 0.37    | 0.09  | 0.99  | 0.90   |
> | OLMo2-7B        | 0.81 ± 0.16    | 0.62  | 1.00  | 0.38   |
> | OLMo2-13B       | 0.58 ± 0.14    | 0.43  | 0.77  | 0.34   |
>
> ---
>
> ### Qwen3 Family
> | Model           | Mean ± Std        | Min    | Max    | Max - Min |
> |-----------------|--------------------|--------|--------|---------|
> | Qwen3-0.6B      | 0.47 ± 0.29    | 0.03  | 0.95  | 0.92   |
> | Qwen3-1.7B      | 0.82 ± 0.10    | 0.69  | 0.96  | 0.27   |
> | Qwen3-4B        | 0.74 ± 0.19    | 0.51  | 0.97  | 0.46   |
> | Qwen3-8B        | 0.82 ± 0.15    | 0.63  | 0.99  | 0.36   |
> | Qwen3-14B       | 0.88 ± 0.13    | 0.68  | 1.00  | 0.32   |
>
> **Datasets.** To examine whether our findings generalize to other types of relations, we extended our experiments to the PEP (Physical Event Plausibility) dataset, which tests commonsense physical reasoning [1]. Using our standard setting (Llama-3.2-3B-Instruct + MEMIT), we found that the same pattern of cross-format fragmentation persists. Editing only with the completion format yields an average efficacy of 0.38 +- 0.18 (min = 0.16, max = 0.70), whereas applying our multi-format supervision (K = 4) substantially improves performance to 0.90 +- 0.07 (min = 0.82, max = 0.98). These results confirm that the **format-conditioned behavior observed in our main experiments also arises in commonsense relational editing**, further reinforcing the generality of the underlying phenomenon.
>
> **Editing Methods.** To ensure that our findings are not tied to a particular method, we additionally evaluated two parametric editing approaches: ROME and AlphaEdit. Tested on $Counterfact_{1k}$ and Llama-3.2-3B-Instruct, the efficacies when edited only with the completion format were: MEMIT (0.64 +- 0.19, min = 0.43, max = 0.91), ROME (0.79 +- 0.22, min = 0.54, max = 1.00) and AlphaEdit (0.70 +- 0.21, min = 0.47, max = 0.97). While the specific performance levels vary across methods, they all exhibit large gaps (> 0.45) between the min and max efficacies. **This confirms that the cross-format divergence we study is prevalent across editing methods.**
>
> [1] Ian Porada, Kaheer Suleman, Adam Trischler, and Jackie Chi Kit Cheung. 2021. Modeling Event Plausibility with Consistent Conceptual Abstraction. In Proceedings of the 2021 Conference of the North American Chapter of the Association for Computational Linguistics: Human Language Technologies, pages 1732–1743, Online. Association for Computational Linguistics.

---

> ### Author Response · Authors · 2025-11-18
>
> ## The contribution of the “distributed storage” concept.
>
> While our analysis focuses on cross-format misalignment in value space, we acknowledge that different formats naturally yield different keys in subject representations. **We conducted ablation studies to isolate the contributions of key variation and value divergence.** We discuss this in the following questions.
>
> We included IKE not to directly support our distributed knowledge subspace hypothesis, but to examine whether generalization failures also arise in non-parametric editing. **Our main focus is on parametric editing, which is more directly tied to the model's representational structure. We clarified this point in the revised version.**
>
> ## Ablation studies to isolate the contributions of key variation and value divergence.
>
> We appreciate the reviewer’s suggestion that variation in the key representations, rather than value-space divergence, might be responsible for cross-format inconsistencies. To disentangle the respective roles of key variation and value divergence, we conducted two targeted ablations: **(i) single-key joint-value editing and (ii) multi-key single-value editing**, where the single-key and single-value corresponds to the completion format. Compared to our multi-key joint-value editing, **the former measures the importance of handling key variation and the latter tests the significance of handling value divergence.**
>
> Our proposed method (multi-key + joint-value) achieves 0.996 +- 0.004 when edited with six formats. When applying single-key joint-value editing, efficacy sharply drops to 0.800 +- 0.065 (min = 0.686, max = 0.861). Conversely, when we apply multi-key single-value editing, performance deteriorates to 0.788 +- 0.213 (min = 0.516, max = 0.997), exhibiting a larger max-min efficacy gap.
>
> These results indicate that **neither key variation nor value divergence alone is sufficient to explain or correct cross-format fragmentation.** Both contribute meaningfully to the observed fragmentation, and **the combination of multi-key access patterns and joint optimization of their corresponding value vectors across multiple formats is essential** for alleviating cross-format divergence. **We incorporated these results in the revised version, specifically in Sec. 4.1.**
>
> ## What if we apply six edits simultaneously, one per format?
>
> We conducted an additional experiment in which we applied six edits simultaneously, one per format, without using our joint loss. This corresponds to treating each format as an independent factual edit under a batch update scheme.
>
> The resulting efficacy was 0.887 +- 0.044 (min = 0.827, max = 0.932), compared to 0.996 +- 0.004 achieved with our joint loss. This indicates that simply batching edits does not resolve the misalignment across format-conditioned subspaces, highlighting the necessity of the joint optimization. **We incorporated these results in the revised version, specifically in Sec. 4.1.**
>
> This phenomenon closely parallels the key conflict problem identified in MEMIT-Merge [2], which reports that MEMIT's efficacy sharply deteriorates when multiple edits share the same subject. The conflict arises when near-identical keys (derived from the same subject entity) are forced to represent different target values, leading to destructive interference across updates. Our multi-format setting naturally induces a related but complementary challenge: whereas the reviewer's question highlights key variation (different keys for the same subject across formats), key conflict represents the opposite extreme, where near-identical keys are associated with multiple values. The proposed joint loss mitigates the key conflict by jointly optimizing all format-conditioned edits to share a coherent update direction in value space.
>
> [2] Zilu Dong, Xiangqing Shen, and Rui Xia. 2025. MEMIT-Merge: Addressing MEMIT’s Key-Value Conflicts in Same-Subject Batch Editing for LLMs. In Findings of the Association for Computational Linguistics: ACL 2025, pages 7952–7960, Vienna, Austria. Association for Computational Linguistics.

---

> ### Author Response · Authors · 2025-11-26
> **Have our clarifications addressed the concerns?**
>
> Dear Reviewer ezRL,
>
> Thank you for your constructive feedback. We have made revisions in response to your comments to clarify and strengthen the contribution of our work. We welcome any additional suggestions.
>
> Please let us know whether the revisions address your concerns.
>
> Thank you again for your time and consideration.

---

### Official Review · Reviewer_4Mj9 · 2025-11-01

**Soundness:** 3
**Presentation:** 2
**Contribution:** 2
**Rating:** 4
**Confidence:** 4

**Summary:**

This paper tackles an important and under-explored problem: how knowledge edits generalize across different task formats (e.g., from completions to multiple-choice questions). The authors find that edits successful in one format often fail to transfer to others, and can even break the model's original consistency. To explain this, they propose the "distributed knowledge storage hypothesis," suggesting that facts are encoded in multiple, format-specific subspaces rather than a single location. They back this up with representational analysis and propose a multi-format supervision method as a potential fix.

**Strengths:**

1. **Novel and Important Problem**: The paper is the first to systematically study generalization across task formats (like completion, MCQA, T/F), moving beyond simple paraphrasing. It clearly highlights that current editing methods fail badly at this.

2. **Novel Hypothesis with Strong Evidence**: The "distributed knowledge storage hypothesis" is an intriguing explanation. It's well-supported by strong representational analysis, like the clustering of edit directions by format (which a linear probe can identify with 98.4% accuracy).

3. **Provides a Practical Solution**: The paper doesn't just identify the problem; it also proposes a practical solution with "multi-format supervision and iterative expansion." It shows this method can efficiently improve cross-format generalization while maintaining specificity.

**Weaknesses:**

1. **Storage vs. Access**: My main concern is the causal leap in the "distributed storage" hypothesis. The evidence (Sec 3.4.2) clearly shows that the edit directions ($v^*-v^0$) are format-specific, but does this necessarily mean the knowledge itself is stored this way? I'm not convinced an alternative isn't just as likely: the knowledge might be stored abstractly, but the access mechanisms (or computational paths) to retrieve it are highly format-dependent, influenced by surface-level prompt differences. The paper needs to grapple with this "storage vs. access" distinction more directly.

2. **Generalizability of the Solution**: The analysis includes IKE, a non-parametric method. Its failure mode might be totally different (e.g., format conflicts in the context window) from MEMIT's. Does the proposed "multi-format supervision" solution even work for IKE? If not, it suggests the fix might be specific to parametric editing, and the scope of the solution should be clarified.

3.  **Interpreting the Consistency Drop**: I'm not sure the drop in consistency is necessarily a bad thing. A high baseline consistency could just mean the model is "consistently wrong." After an edit, it becomes "inconsistently right" (correct in one format, wrong in others), causing the score to drop. This seems like an artifact of a partially successful edit. Relatedly, I don't see why erasing the frequency bonus strongly points to distributed storage. It seems just as compatible with the "distributed access" idea (i.e., high-frequency facts have more access paths, and the edit only broke one).

**Questions:**

See weaknesses

---

> ### Author Response · Authors · 2025-11-18
>
> Thank you for the constructive feedback.
>
> ## Storage vs. Access
>
> Following prior work (e.g., ROME, MEMIT), we initially treated the MLP of the last subject token as the knowledge storage location, with subsequent attention layers performing knowledge access. We agree, however, that in end-to-end models, the boundary between storage and access is inherently blurred. **To avoid implying a strict causal distinction between storage and access, we adopt the term 'distributed knowledge subspace hypothesis' instead of 'distributed storage'. We have incorporated this into the revised manuscript.**
>
> ## Scope of the solution
>
> We included IKE not to directly support our distributed knowledge subspace hypothesis, but to examine whether generalization failures also arise in non-parametric editing. **Our main focus is on parametric editing, which is more directly tied to the model's representational structure. We clarified this point in the revised version.**
>
> ## Interpreting the consistency drop
>
> We agree that a lower post-edit consistency score does not necessarily indicate a degradation in model quality. Our analysis does not treat the drop as intrinsically negative but as a signal that the effects of an edit are distributed unevenly across formats. The purpose of this metric is to reveal cross-format divergence, not to penalize partial success. Importantly, we view efficacy and consistency as complementary objectives, and the ultimate goal of robust knowledge editing is to achieve consistently right predictions.
>
> Regarding the frequency-bonus experiment, we agree that high-frequency facts may have more access paths and the edit only affected a subset. **The key point here is that high frequency in training data does not ensure learning a shared subspace across multiple forms for the same fact. In the revised version, we adjusted the consistency-related discussion to better highlight this key point.** Again, robust knowledge editing aims for consistently right edits, which may be achieved either by (i) influencing all pathway-specific subspaces associated with that fact, or (ii) modifying a more abstracted, shared subspace that multiple pathways converge on.

---

> ### Author Response · Authors · 2025-11-26
> **Have our clarifications addressed the concerns?**
>
> Dear Reviewer 4Mj9,
>
> Thank you for your constructive feedback. We have made revisions in response to your comments to clarify and strengthen the presentation of our work. We welcome any additional suggestions.
>
> Please let us know whether the revisions address your concerns.
>
> Thank you again for your time and consideration.

---

### Author Response · Authors · 2025-12-01
**Post-discussion Summary**

Dear AC and Reviewers,

We sincerely appreciate the thoughtful and detailed feedback provided during the review and discussion period. Below we summarize the key strengths highlighted by the reviewers and the major revisions we have implemented in response.

Strengths noted by the reviewers include:
- The paper is the first to systematically study generalization across task formats (completion, MCQA, TF/YN, ODQA), **filling a missing dimension in current knowledge-editing benchmarks**.
- The proposed distributed knowledge subspace **hypothesis is novel**, and strongly **supported by empirical and representational evidence**.
- The multi-format supervision with iterative expansion method is **simple, practical, and consistently improves cross-format generalization** while maintaining specificity.

In response to the reviewers' feedback, we have substantially strengthened the paper along the following key dimensions:
- **Expanded experimental scope:**
We added extensive evaluations across 10 models that cover three model families (Llama, OLMo2, Qwen3) and larger parameter scales (0.6B-14B), a new dataset (PEP), and additional editing baselines (ROME, AlphaEdit). **We consistently observed cross-format divergence across diverse models, datasets, and editing methods.**
- **Critical ablations added:**
**Both key variation and value divergence contribute to cross-format failure, and the proposed method effectively addresses both** with multi-key joint-value.
- **Improved analysis of iterative expansion**
- **Clarified conceptual claims**

We believe these revisions significantly enhance the clarity, rigor, and generality of our contributions.

Thank you again for your time and effort in reviewing our submission.

Best regards,
Authors

---

### Meta-Review · Area_Chair_XQ6D · 2025-12-30

**Summary:**

The following four types of concerns can be summarized below:

(1) Model Limitation: The study only validates results on LLaMA-3.2-3B-Instruct, lacking experiments on larger parameter models (e.g., 70B, GPT) or alternative architectures (e.g., DeepSeek, Qwen), making it difficult to demonstrate the generalizability of the conclusions.

(2) Dataset Limitation: Only CounterFact-1k is used, with no validation across multilingual data, fine-tuned models, or other knowledge domains (e.g., commonsense reasoning, multi-hop inference, temporal knowledge).

(3) Storage vs. Access: Editing failures across different formats might not indicate that knowledge is "distributed in storage", but rather that the access mechanisms differ.

(4) Explanation of Consistency Degradation: The observed decline in consistency may simply be an artifact of partial editing success, rather than evidence of the underlying storage mechanism.
Shallow Agent Analysis: The evaluation of agent-assisted generation lacks quantitative analysis (e.g., in terms of quality and diversity).

**Reviewer Concerns:**

The reviewer's concerns may be addressed by the rebuttal:

The authors have added experiments on different datasets and introduced new model baselines. They have also provided a more detailed analysis of the agent component. Therefore, I consider the concerns regarding Model Limitation, Dataset Limitation, and Shallow Agent Analysis to be addressed. The analysis of the Consistency metric has also been completed.

The reviewer's concerns may still be outstanding:

Reviewer ezRL points out in W2 that the contribution of distributed storage is limited. The question states that editing knowledge in different formats is essentially the same as the method proposed by the authors and can achieve the same result. Although the authors provide empirical evidence showing their method performs better, the comparison only considers a single-edit strategy for knowledge in different formats.

**Reviewer Scores:**

For reviewers 4Mj9 and ezRL, their ratings are unlikely to improve because the Storage vs. Access issue remains unresolved.

For reviewer U1ok, his ratings may improve because his questions have been fully answered.

For reviewer 1i3Q, he has confirmed that their issue has been resolved, which will likely improve their ratings.

---

### Decision · Program_Chairs · 2026-01-26

Reject